# OpenReview forum: "Defending LVLMs Against Vision Attacks Through Partial-Perception Supervision"
_ICML.cc/2025/Conference — ICML 2025 poster_

### Official Review · Reviewer_dVJV · 2025-02-22

**Overall Recommendation:** 3

**Summary:**

The authors address the vulnerability of LVLMs to adversarially perturbed or maliciously injected input images. Their motivation is to solve the issues of response quality degradion that comes with the existing defense methods which rely on image modifications that could possibly cause semantic distortions. They propose **DPS (Defense through Partial-Perception Supervision)**, a novel black-box and training-free defense mechanism. Instead of directly using responses from partial images for voting, DPS employs these responses to guide the LVLM’s inference on the original image. This allows the model to adjust its response when under attack while preserving accuracy on clean inputs. They also demonstrate the effectiveness of DPS, and claim to reduce the average attack success rate by 76.3% across six datasets and three widely used LVLMs, outperforming baseline defenses.  In essence, DPS leverages a “weak-to-strong” learning strategy where partial, and potentially less accurate, observations help correct and stabilize the full-model output in the face of malicious input perturbations.

**Claims And Evidence:**

The experimental results and ablation studies presented in the paper provide strong empirical backing for the claims. The authors demonstrate that DPS can reduce the average attack success rate by approximately 76.3% across multiple datasets and LVLMs, while also maintaining standard performance on clean inputs. Comprehensive comparisons with baseline defenses and detailed evaluations (including different cropping strategies and safety metrics) support the effectiveness of the proposed method. Therefore, the claims made in the submission are indeed supported by clear and convincing empirical evidence.

**Essential References Not Discussed:**

While the authors aim to improve response quality while reducing attack success rates, the paper overlooks recent literature that tackles this problem through detection-based approaches. The following two papers are particularly relevant and should be considered:

1. **VLMGuard: Defending VLMs against Malicious Prompts via Unlabeled Data** (Du et al., 2024) – This work introduces a defense mechanism that leverages unlabeled data to detect and mitigate adversarial prompts targeting VLMs. Unlike DPS, which focuses on partial perception, VLMGuard explores input-space robustness via prompt-level interventions.

2. **MirrorCheck: Efficient adversarial defense for vision-language models** (Fares et al., 2024) – This approach focuses on efficient adversarial defenses through self-consistency checks within multimodal inputs. MirrorCheck employs consistency-based detection to identify manipulated inputs, providing an alternative or complementary strategy to DPS.

These works introduce alternative ways to achieve robustness in LVLMs that could complement or contrast with DPS. The authors should explore these and other recent efforts in adversarial defenses for a more comprehensive discussion.

**Experimental Designs Or Analyses:**

The experimental design was thoroughly examined, including evaluations on misleading attacks, jailbreak scenarios, and standard performance (using the MM-Vet benchmark). The authors compare DPS against multiple baseline defenses across six datasets and three LVLMs, and they conduct detailed ablation studies (e.g., comparing different cropping strategies) to validate the design. Overall, these analyses appear sound and well-constructed. A couple of points could be addressed to strengthen the work further:
- The computational overhead introduced by multiple cropping strategies and the multi-agent interaction in DPS might affect scalability in practical deployments.
- Including additional details on statistical significance or error analysis (e.g., confidence intervals or p-values) would enhance the clarity and rigor of the empirical results.

Nonetheless, the experimental design and analyses are valid and effectively support the authors’ claims.

**Methods And Evaluation Criteria:**

Yes, the proposed method and evaluation criteria are well-suited for the problem. DPS leverages partial observations to guide the full-model responses, addressing both attack mitigation and standard performance. Additionally, the use of multiple benchmark datasets and comparisons with established baseline methods provides a robust and comprehensive evaluation framework, ensuring that the method’s effectiveness is thoroughly validated.

**Other Comments Or Suggestions:**

Please refer to the strengths and weaknesses listed above for a comprehensive assessment.  Furthermore, it would be beneficial for the authors to include a discussion comparing DPS with VLMGuard and MirrorCheck. For instance, while DPS leverages partial perception supervision to guide the model's full-image response under attack, VLMGuard utilizes unlabeled data to detect malicious prompts, and MirrorCheck employs self-consistency checks between modalities. A direct empirical or high-level comparison could further clarify the relative strengths and limitations of these approaches.

**Other Strengths And Weaknesses:**

- The presentation and clarity of the approach are commendable. The method is clearly described and the experimental results demonstrate a significant empirical improvement over prior works.
- While the idea is novel in its application of partial perception supervision to mitigate adversarial attacks, its conceptual foundation is similar to existing literature, positioning it as an incremental improvement.
- The paper would benefit from an exploration of failure cases. Specifically, it is unclear how DPS handles scenarios where the supervision mechanism is bypassed or when a jailbreak input goes undetected, areas where detection-based methods might offer more reliable performance.
- A discussion of edge cases, potential weaknesses, and failure modes would enhance the robustness and clarity of the paper.
- Finally, providing a theoretical analysis or high-level intuition for why DPS improves robustness over prior methods would further strengthen the contribution and support its practical applicability.

Overall, great work by the authors, with room for improvement in addressing limitations and offering deeper theoretical insights.

**Questions For Authors:**

Please check strengths and weaknesses above.

**Relation To Broader Scientific Literature:**

The key contributions are well stated, and the methodology is necessary to tackle both alignment challenges and quality issues found in existing solutions. Prior defenses rely on majority voting across modified images, which degrades response quality, but this paper builds on this by using the modified images to supervise the model’s inference, maintaining accuracy on clean inputs while reducing attack success rates. This builds on existing research in robust AI alignment by showing that adversarial robustness can be improved without retraining. The results make a strong case for DPS as a practical and effective defense, adding to ongoing discussions on safe and trustworthy AI systems.

**Theoretical Claims:**

There are no formal proofs or theoretical claims in the paper. The authors primarily support their approach with intuitive reasoning and robust empirical evidence. While the intuition behind DPS and its ability to improve robustness is compelling, a dedicated section (perhaps in the appendix) that delves into theoretical insights or offers a more formal justification of why DPS improves robustness would further strengthen the paper.

---

> ### Author Rebuttal · Authors · 2025-04-01
>
> We greatly appreciate your time and effort in reviewing our work and would like to address each of your concerns.
> ### 1. The incremental improvement
> We appreciate the reviewer’s recognition of our idea novelty. In addition to the novel idea, we respectfully argue that the reviewer should reconsider the multifaceted and fundamental contributions in this work：
> - We provide significant insights into LVLM defense. We identify a key vulnerability pattern in LVLMs: reduced confidence with attacked images versus high confidence with clean inputs (Sections 3.2-3.3).
> - The training-free deployment and superior defense performance demonstrate DPS's practical distinctiveness.
> - Our systematic bridging of "weak-to-strong" principles with LVLM defense mechanisms creates a new research nexus.
>
> More contribution details are in *Reviewer 5ot1 [2. Suitable for the ICML community]*. These interlocking advances collectively represent a non-incremental leap in defense, which we will further clarify.
> ### 2. Theoretical Justification of DPS
> We provide a theoretical formulation to explain why our work could achieve defense.
> 1) The Target of Attack
>
> Let $x$ denote the original example, $x' = \text{adv}(x, \delta)$ denote the adversarial example, and $F_v(x) \in \mathbb{R}^d$ be its visual features extracted by the vision encoder. $T_\text{target}$ is the targeted output, $T_\text{clean}$ is the originial output of clean input, and $\mathcal{T}$ represents the text space. We can represent the probability of $T_{\text{clean}}$ and $T_{\text{target}}$ condition on the $x$ or $x'$:
> $$
> P(T_{\text{clean}}|x') = \frac{P(T_{\text{clean}}|F_v(x'))}{\sum_{T\in\mathcal{T}} P(T|F_v(x'))}.
> $$
> After attack, $P(T_{\text{target}} | x')$ becomes high and the  $P(T_{\text{clean}} | x')$ is largely reduced.
>
> 2) Prior Injection via Partial-Perception Supervision
>
> Our method DPS crops the original image and generates text $T_c$ when we feed the partial images into LVLM. We have $\phi(T, T_c) \in [0, 1]$ representing the semantic consistency between the candidate text $T\in \mathcal{T}$ and $T_c$.
> Compared with the attack formulation, our defense mechanism actually introduces crop-induced text $T_c$ as a **prior** to constrain the output distribution. The revised conditional probability of the defense can be approximated as:
> $$
> P_{\text{defense}}(T_\text{clean}|x',T_c) = \frac{P(T_\text{clean}|F_v(x')) \cdot \phi(T_\text{clean}, T_c)}{\sum_{T\in\mathcal{T}} P(T|F_v(x') \cdot \phi(T, T_c)}.
> $$
> As the $ \phi(T_\text{clean}, T_c)$ is relatively high compared with $ \phi(T_\text{target}, T_c)$, we can have $P_{\text{defense}}(T_\text{clean}|x',T_c)>P_{\text{defense}}(T_\text{target}|x',T_c)$, achieving defense.
> ### 3. Computational overhead
> The computational overhead of DPS yields proportionally higher defense effectiveness, while existing methods failed to improve defensive capability even with the increased computation budgets. Please refer to our response to *Reviewer popy regarding [1. Computational Cost]*.
> ### 4. Statistical significance
> Thanks for your suggestion! Please refer to our response to *Reviewer 2wMM regarding [1. Error Bar]*.
> ### 5. Comparison with more recent work
> We sincerely appreciate the relevant literature you shared and will incorporate these papers into the revised manuscript.
> DPS fundamentally differs from the two detection methods as follows:
> - DPS is a **training-free** defense that offers significantly lower deployment costs in wild.
> - The scopes of attacks are different.  These two methods are limited to specific scenarios (i.e., jailbreak detection) and **cannot be directly applied to the defense against other attacks like misleading attacks**.
> - The underlying mechanisms are fundamentally distinct. DPS leverages the sensitivity of diverse attacks to cropping operations, achieving defense.
> Further, we conduct an empirical study to compare the performance between DPS and MirrorCheck on MM-Safety and VisualAttack datasets.
>
> From Table 1, we observe that while MirrorCheck's defense performance is only slightly worse than our method. Its standard performance is significantly lower than that of LS-DPS due to the high false positive rate.
> Since VLMGuard's implementation is not publicly available, we will revisit this comparison with more time.
>
> Table 1. Defense Performance Compared with MirrorCheck
> |  | ASR (MM-Safety) | ASR (VisualAttack) | Avg. Standard Performance (MM-Vet) $\uparrow$|
> | -------- | -------- | -------- | -------- |
> | MirrorCheck   |  0.20  |  0.17 | 0.26
> | LS-DPS     |   0.10   |   0.17 |  0.68
>
> ### 6. An exploration of failure cases.
> The most common failure case occurs when the cropping fails to eliminate attacks, rendering partial perception supervision ineffective for defense. In such cases, combining detection-based defense methods may enhance defense effectiveness, an area for future exploration.

---

### Official Review · Reviewer_5ot1 · 2025-03-10

**Overall Recommendation:** 3

**Summary:**

This paper proposes a black-box, training-free defense method called DPS (Defense via Partial Perception Supervision) to protect large vision-language models (LVLMs) from visual adversarial attacks. Unlike existing defenses (e.g., majority voting), which suffer from semantic distortion due to image cropping and thus degrade performance on clean inputs, DPS leverages “weak supervision” responses generated from cropped image regions to dynamically guide the model’s predictions during inference. When an adversarial input is detected, partial perception information encourages the model to lower confidence and correct erroneous responses, while for clean inputs, the model maintains its original high-confidence predictions. Inspired by the “weak-to-strong learning” paradigm, DPS treats partial perception as weak supervision guiding a stronger model. It employs a multi-cropping strategy to generate supervisory signals and incorporates safety prompts to enhance robustness. Experiments demonstrate that DPS reduces the attack success rate by an average of 76.3% across six datasets, outperforming baseline methods in defending against misleading and jailbreak attacks while preserving performance on standard vision-language tasks (MM-Vet benchmark), showcasing its effectiveness and practicality against adversarial visual attacks.

**Claims And Evidence:**

The paper claims that LVLMs may exhibit a decrease in confidence when processing perturbed adversarial images. However, I am skeptical about this confidence variation because the definition of confidence in this context is unclear. Specifically, I question whether this confidence is inferred based on the textual responses of LVLMs or directly derived from the model’s output logits. A more detailed explanation is needed to clarify this aspect.

**Essential References Not Discussed:**

Related work has been discussed.

**Experimental Designs Or Analyses:**

This paper conducts experiments on multiple datasets and tasks, making the evaluation relatively comprehensive. However, I still have some questions:

1. How is the adversarial noise for images generated? It is well known that performing targeted white-box attacks on LVLMs is extremely challenging and typically results in very low attack success rates. If the adversarial examples are optimized using an open-source LVLM model, it is unlikely that they will effectively transfer to closed-source models.

2. Although evaluating some closed-source models enhances the generalizability of the proposed method, for the sake of reproducibility, it is necessary to assess its effectiveness on an open-source model, such as LLaVA or Qwen2-VL.

**Methods And Evaluation Criteria:**

Overall, this method appears to be simple and reasonable. However, such a straightforward approach may offer limited insights to the field.
Therefore, I have some doubts about whether this work is suitable for ICML. I’m not entirely certain, but I feel that this paper might be a better fit for ACL.

**Other Comments Or Suggestions:**

See above

**Other Strengths And Weaknesses:**

See above

**Questions For Authors:**

See above

**Relation To Broader Scientific Literature:**

This issue is meaningful for the security of multimodal large models.

**Theoretical Claims:**

The submission ​does not present formal theoretical claims or proofs, as it is primarily an empirical study focused on methodology and experimental validation.

---

> ### Author Rebuttal · Authors · 2025-04-01
>
> We greatly appreciate your time and effort in reviewing our work, and would like to address each of your concerns.
>
> ### 1. The definition of confidence
> > However, I am skeptical about this confidence variation because the definition of confidence in this context is unclear.  Specifically, I question whether this confidence is inferred based on the textual responses of LVLMs or directly derived from the model’s output logits. A more detailed explanation is needed to clarify this aspect.
>
> Thank you for the comments. In our work, we define LVLM models' confidence as the inverse of the standard deviation of responses to variations of the same question, crafted by adding different prefixes to the original question. This confidence metric represents certainty in model outputs without requiring access to internal model parameters. A high confidence value indicates a low standard deviation and, thus, low uncertainty in the LVLM's responses. To avoid potential ambiguity in the term 'confidence,' we will add clear explanations of the term in the revised manuscript.
>
> ### 2. Suitable for the ICML community
> >  However, such a straightforward approach may offer limited insights into the field. Therefore, I have some doubts about whether this work is suitable for ICML. I’m not entirely certain, but I feel that this paper might be a better fit for ACL.
>
> We respectfully disagree with the assessment that our paper might be more suitable for ACL due to its "straightforward approach."
>
> - First, **Novel insights for LVLM Defense**. Our paper provides significant insights into LVLM defense: (1). We identify a key vulnerability pattern in LVLMs: reduced confidence with attacked images versus high confidence with clean inputs (Sections 3.2-3.3). (2). We innovatively apply the "weak-to-strong" learning phenomenon to visual defense by using partial image perception to supervise full-image understanding. (3) Our approach fundamentally differs from existing methods that rely on direct majority voting or filtering.
> - Second, **Demonstrated Effectiveness**. (1). Our method reduced attack success rates by 76.3% across six datasets on three models (Section 5.2). (2). We show effectiveness against both misleading and jailbreak attacks (Tables 1-2). (3) Unlike other methods (e.g., SmoothVLM), we maintained standard performance while improving defense (Section 5.3).
> - Third, **Simplicity as Strength**. ICML values impact and effectiveness, not complexity for its own sake. Our training-free method requires no model modification, making it immediately applicable to any LVLM. The DPS framework shows strong compatibility with other defense strategies (Section 4.4).
>
> Our paper presents novel insights, effective solutions, and provides new solutions in defending LVLMs, aligning well with ICML's emphasis on machine learning advances.
>
>
> ### 3. How is the adversarial noise for images generated?
> > It is well known that performing targeted white-box attacks on LVLMs is extremely challenging and typically results in very low attack success rates. If the adversarial examples are optimized using an open-source LVLM model, it is unlikely that they will effectively transfer to closed-source models.
>
> In our experiments, in addition to the adversarial samples from existing datasets (RTA 100, MultiTrust, SafetyBench, and HADES), we generate adversarial samples using the VisualAttack method (lines 250-266). These samples achieve a transfer success rate of approximately 10% to 20%. The samples selected for defense evaluation are those that demonstrate successful transferability.
>
> ### 4. Experiments on open-source models
> > Although evaluating some closed-source models enhances the generalizability of the proposed method, for the sake of reproducibility, it is necessary to assess its effectiveness on an open-source model, such as LLaVA or Qwen2-VL.
>
> Thanks for your suggestion! We conduct extended experiments using the open-source model Qwen2.5-VL-32B on misleading attack datasets—RTA 100, Self-Gen, and MultiTrust—as well as jailbreak datasets MM-Safety and VisionAttack. The results (ASR) of misleading defense and jailbreak defense are shown in Table 1 and Table 2, while the standard performance is shown in the Figure (https://anonymous.4open.science/r/ICML2025-231D/README.md). The proposed methods, DPS and LS-DPS, consistently achieve the best performance against misleading and jailbreak attacks.

---

### Official Review · Reviewer_2wMM · 2025-03-14

**Overall Recommendation:** 4

**Summary:**

The paper proposes a improved ensemble based defense of VLMs. The idea improves upon SmoothVLM which leveraged partial crops or random perturbations of the input image and ensembled the responses, by adding a secondary step of using the VLM responses on the partial crops as supervisory inputs to the VLM. The approach also provides an optional language post-processing step that converts malicious responses to benign. The authors evaluate this defense on several jail breaking and adversarial datasets, and show that the approach is succesful at defending against jail-breaking attacks.

## Update after rebuttal

The rebuttal has mostly answered my queries. I am keeping my score, with the suggestion that the authors add error bars to the reported metrics.

**Claims And Evidence:**

The claims in the paper are well-supported by the analysis.

**Essential References Not Discussed:**

The paper covers necessary related work.

**Experimental Designs Or Analyses:**

I have checked all the experimental designs and analyses and believe them to be generally sound. It would be additionally useful to analyse the effect of the number of partial crops used in the defense.

**Methods And Evaluation Criteria:**

I find the method to be very intuitive, and appreciate the fact that they are able to achieve training free robustness. The experiments are well-designed and the evaluations are thorough. The authors provide a comprehensive evaluation over two adversarial and one benign tasks, as well as analyse the effectiveness the cropping strategy. However, I would suggest adding error bars to the results where ever possible to account for the random nature of the evaluations.

**Other Comments Or Suggestions:**

None

**Other Strengths And Weaknesses:**

The approach is quite well motivated and original. It provides a training-free approach towards improved robustness. I appreciate the simplicity of the approach itself.

**Questions For Authors:**

I do not have any specific questions However, as suggested above, I would like to see the effect of number of evaluations on adversarial robustness.

**Relation To Broader Scientific Literature:**

The paper is related to the broader scientific literature on defending against adversarial attacks on VLMs. The closest prior work is SmoothVLM which uses a similar ensemble based defense. The proposed approach improves upon SmoothVLM using the partial input responses as supervisory signals to the VLM insted of as inputs to an ensemble.

**Theoretical Claims:**

Not applicable.

---

> ### Author Rebuttal · Authors · 2025-04-01
>
> We greatly appreciate your time and effort in reviewing our work, and would like to address each of your concerns.
>
> ### 1. Error Bar
> > I would suggest adding error bars to the results where ever possible to account for the random nature of the evaluations.
>
> Thanks for your suggestion! We conducted all our experiments three times to calculate the error bars and incorporated them into the results. Moreover, we extend our experiments to the open-source model Qwen2.5-VL-32B. Here we only show the error bars for Qwen2.5-VL-32B. (https://anonymous.4open.science/r/ICML2025-231D/README.md)
>
> In Table 1, we demonstrate the statistical significance using a t-test, with p-values consistently less than 0.05, confirming their significance. Here, the t-test is performed between DPS and the baseline with the best performance.
>
> Table 1. P-values from the t-test
> | Dataset | RTA 100 | Self-Gen | MultiTrust | MM-safety |VisAttack |
> | -------- | -------- | -------- |-------- |-------- |-------- |
> | p-values     |  p=0.03   |   p=0.02  | p=0.03   | p=0.04  | p=0.03   |
>
>
> ### 2. Analysis of the number of partial crops
> > It would be additionally useful to analyse the effect of the number of partial crops used in the defense.
>
> Thanks for your suggestion! Due to time limitations, we restrict our analysis to random crop-based partial copies (varying from 1 to 5) to evaluate their impact on defense performance, using the Self-Gen dataset. The experimental results in Table 2 demonstrate a positive correlation between the number of partial crops and defense performance.
>
> Table 2. Attack Success Rate (ASR) Comparison Under Different Partial Crop Numbers
> |  Crop Numbers | 1 |  2 | 3 | 4 | 5|
> | -------- | -------- | -------- |-------- |-------- |-------- |
> |   Self-Gen   | 0.55 |  0.30    | 0.30 |  0.30 | 0.25

---

### Official Review · Reviewer_popy · 2025-03-14

**Overall Recommendation:** 2

**Summary:**

This paper proposes a novel method, named Defense through Partial Perception Supervision (DPS), which focuses on evaluating and improving the robustness of Large Vision-Language Models (LVLMs) against vision attacks. Specifically, DPS leverage the outputs from cropped image processing to supervise the outputs from full image processing, to provide a correct understanding of images under vision attacks. Empirical experiments show DPS outperforms the baseline methods across six datasets and three LVLMs.

**Claims And Evidence:**

Yes.

**Essential References Not Discussed:**

No.

**Experimental Designs Or Analyses:**

Please refer to ‘Questions For Authors’.

**Methods And Evaluation Criteria:**

Yes.

**Other Comments Or Suggestions:**

1. To enhance the clarity and structure of the manuscript, it is suggested to outline the specific contributions of this work in a bullet-point format in the Introduction section
2. A list of typos. Line 186, it should be 'There are no animals.' Line 303, 'employe' should be corrected to 'employ.' Line 354, In 'Specifically,' the following 'the' should be in lowercase."

**Other Strengths And Weaknesses:**

Strengths.
1. The paper is well-organized and easy to follow.
2. The paper conducts extensive experiments, comparing with a wide range of defense methods on various datasets, thoroughly validating the effectiveness of the proposed approach.

Weaknesses.
1. Complexity and computational cost. For DPS, it provides supervision for the full model by constructing multiple part-perception supervisions composed of various cropping methods, such as center cropping, random cropping, and adaptive cropping. Compared to other SOTA methods, the multiple query mechanism inevitably increases the time overhead during the inference phase. According to the results in Section C.3, the efficiency of LS-DPS is only superior to Smoothvlm, and the efficiency results for IVA and Warning are missing in Section C.3.

2. Clarity. This paper draws an analogy, treating responses to partial images as those from weak models and responses to full images as those from strong models. Is this hypothesis, or analogy, reasonable? According to the methodology, Part-Perc models and full models are essentially the same model, except that they process partial images and full images, respectively. Therefore, I am more inclined to view this as information fusion across different levels rather than a form of weak-to-strong learning.

3. Soundness. I suppose that DPS is not entirely black-box. As shown in Figure 2, the original question specifically emphasizes whether there are animals in the image, which leads to information leakage. In other words, the mention of 'animal' in the original question aligns with the attack target of the adversarial attack, which significantly helps LVLMs focus on the animals in the image. If the original question only asks what is contained in the image without emphasizing animals, I believe the ASR would increase substantially. In other words, under supervision without specific content constraints, the second answer provided by the full model should randomly deviate from the original answer.

**Questions For Authors:**

1. I am curious about the effectiveness of traditional adversarial purification techniques from deep learning models in defending LVLMs against vision-based attacks. To my understanding, adversarial purification can mitigate adversarial perturbations through two main approaches. The first approaches [1,2,3] involve post-processing methods, such as denoising, compression, total variation minimization, and image quilting, aiming to eliminate adversarial noise. The second approaches [4,5,6] leverage generative models like GANs or diffusion models to reconstruct a clean sample from the adversarial input. Adversarial purification method can achieve similar performance to DPS but with significantly higher efficiency. Unlike DPS, which requires multiple interactions with the large model to refine responses, adversarial purification only needs to process the adversarial sample once before passing it to the model for evaluation. This streamlined process makes it a more practical and efficient defense mechanism.

[1]	Guo C, Rana M, Cisse M, et al. Countering adversarial images using input transformations. ICLR. 2018.
[2]	Xie C, Wang J, Zhang Z, et al. Mitigating adversarial effects through randomization. ICLR. 2018.
[3]	Xu W, Evans D, Qi Y. Feature squeezing: Detecting adversarial examples in deep neural networks. NDSS. 2018.
[4]	Jin G, Shen S, Zhang D, et al. Ape-gan: Adversarial perturbation elimination with gan. ICASSP. 2019:
[5] Nie, W, Guo, Brandon, Huang, Y, et al. Diffusion Models for Adversarial Purification. ICML. 2022.
[6] Lee M, Kim D. Robust evaluation of diffusion-based adversarial purification. ICCV. 2023.


2. In Section 4, is the cropping methods, such as center cropping, random cropping, and adaptive cropping, the only way to obtain partial images? If so, please explain the underlying reasons. If not, please describe other feasible operations.

3. For the caption of Table 1, I find the use of the term ‘adversarial samples’ unclear. To my understanding, ‘adversarial samples’ specifically denote samples which are intentionally modified with imperceptible perturbations to deceive Deep Neural Networks. However, since the datasets referenced in Table 1 do not include VisualAttack or any adversarial manipulation, the presence of ‘adversarial samples’ in this context seems inconsistent and potentially misleading

4. In Table 2 (line 337), the results for Qwen-VL-Plus's adversarial defense against VisualAtt show a notable discrepancy between DPS and LS-DPS. Considering the discussion in Section 4.4, where only an additional prompt was introduced, the significant improvement in adversarial defense performance appears unexpected. Even if the model classifies the adversarial attack target as 'harmful,' such a substantial enhancement in defense effectiveness seems unlikely and warrants further explanation.

5. In the Implementation Details, please elaborate on the specific methodology behind adaptive cropping (AC). As evidenced in Table 3, AC consistently outperforms other cropping techniques across most datasets in terms of ASR. For datasets such as Self-Gen, MultiTrust, and MM-Safety, is the observed improvement in defense performance attributed to AC’s ability to directly eliminate textual contents through cropping?

6. In Section 5.3, as illustrated in Figure 4 and Table 4, why does the Warning Prompt significantly enhance the standard performance of Qwen-VL-Plus? Please provide a concrete example to explain the underlying reason for this improvement

**Relation To Broader Scientific Literature:**

On one hand, this paper utilizes cropping operations to reduce the interference caused by vision attacks on images, which aligns with the idea of adversarial purification [1,2,3,4,5,6] in conventional adversarial defense. Both methodologies capitalize on the inherent sensitivity of adversarial perturbations to modifications like smoothing, compression, and denoising. On the other hand, the dual-stage framework enhances the robustness of LVLMs by transitioning from partial to full perception, which shares certain parallels with the concept of weak-to-strong learning [7,8], although it is not entirely analogous.


[1]	Guo C, Rana M, Cisse M, et al. Countering adversarial images using input transformations. ICLR. 2018.
[2]	Xie C, Wang J, Zhang Z, et al. Mitigating adversarial effects through randomization. ICLR. 2018.
[3]	Xu W, Evans D, Qi Y. Feature squeezing: Detecting adversarial examples in deep neural networks. NDSS. 2018.
[4]	Jin G, Shen S, Zhang D, et al. Ape-gan: Adversarial perturbation elimination with gan. ICASSP. 2019:
[5] Nie, W, Guo, Brandon, Huang, Y, et al. Diffusion Models for Adversarial Purification. ICML. 2022.
[6] Lee M, Kim D. Robust evaluation of diffusion-based adversarial purification. ICCV. 2023.
[7] Khan, A., Hughes, J., Valentine, D, et al. Debating with more persuasive llms leads to more truthful answers, 2024. URL https://arxiv.org/abs/2402.06782.
[8] Yang, Y., Ma, Y., and Liu, P. Weak-to-strong reasoning, 2024. URL https://arxiv.org/abs/2407.13647.

**Theoretical Claims:**

This paper draws an analogy, treating responses to partial images as those from weak models and responses to full images as those from strong models. Is this hypothesis, or analogy, reasonable? According to the methodology, Part-Perc models and full models are essentially the same model, except that they process partial images and full images, respectively. Therefore, I am more inclined to view this as information fusion across different levels rather than a form of weak-to-strong learning.

---

> ### Author Rebuttal · Authors · 2025-04-01
>
> We greatly appreciate your effort and address each of your concerns.
>
> ### 1. Computational cost
>
> We report the computational costs of baselines and our method, and would like to highlight that the computational overhead of our DPS delivers proportionally higher defense effectiveness. Specifically, for a fair comparison, we enhanced the baselines by running each model 6 times and implementing majority voting to achieve ensemble effects comparable to our method's six queries. As shown in Table 1 (https://anonymous.4open.science/r/ICML2025-1-3F36/README.md), even with additional inference time, the enhanced baselines cannot achieve proportional improvements in defensive effectiveness comparable to DPS.
>
> ### 2. Why use the "weak-to-strong" analogy
>
> In the ICML 2024 best paper [Khan; ICML 2024], they establish a framework where an LLM without certain necessary information serves as the 'weak model', while the same LLM with full information access functions as the 'strong model.' Inspired by this, we apply a similar analogy—treating models with partial-image access as 'weak' variants to enhance 'strong' full-image models. Our black-box, training-free method leverages this approach across six attack datasets (lines 43-53).
> While this could be viewed as information fusion, we believe the 'weak-to-strong' analogy particularly suits our method and will clarify this in our revision.
>
> ### 3. Black-box nature of our method
>
> Our method operates in a fully black-box manner—the defense process works independently of questions. In Figure 2, questions aren't involved in our defense method.
> Regarding concerns about "animals" keywords causing information leakage:
> Jailbreak benchmark prompts contain no such keywords, yet our method performs well.
> Replacing "animals" with "things" still triggers attack-related content, and removing animal-related statements entirely doesn't affect output accuracy.
> We'll revise Figure 2 for clarity.
>
> ### 4. Comparing with adversarial purification.
>
> Although DPS can also be viewed as a pre-processing approach, it fundamentally differs from adversarial purification:
>
> - First, the scopes of attacks are different. Adversarial purification mainly focuses on removing pixel-wise adversarial noise (e.g., DiffPure [Nie; ICML 2022]). However, non-noise attacks like typographic attacks [Cheng; ECCV 2024] and jailbreak attacks [Shayegani; ICLR 2024] cannot be handled effectively by adversarial purification. In contrast, our DPS counteracts both adversarial noise and typographic attacks.
> - Second, the underlying mechanisms are fundamentally distinct. Adversarial purification removes or breaks the effectiveness of adversarial noise. In contrast, our approach leverages the sensitivity of diverse attacks to cropping operations, naturally defending against non-noise attacks like the misleading attacks in our submission. As shown in Tables 2-3 (https://anonymous.4open.science/r/ICML2025-1-3F36/README.md), our DPS outperforms all purification baselines under both misleading and jailbreak attacks.
>
> ### 5. Why do we use cropping?
>
> We use center cropping, random cropping, and adaptive cropping methods for these reasons:
>
> - Efficiency: Cropping is highly concise with nearly no deployment cost.
> - Effectiveness: Cropping effectively eliminates various attacks, including adversarial noises and typographic attacks.
>
> While advanced models like SAM [Kirillov; ICCV 2023] could be used, our tests showed higher costs without significant improvements. We'll explore more effective cropping operations in future work.
>
> ### 6. Defense enhancement on Qwen-VL-Plus
>
> The primary reason for the significant enhancement is that Qwen-VL-Plus's inherent safety mechanisms (which DPS relies solely on) are substantially weaker than those of the LLM-based safety checker (i.e., GPT-4o-Mini in our paper), which complements DPS in LS-DPS. In contrast, other LVLMs, whose safety mechanisms are closer to those of LLMs, exhibit less pronounced improvements. This aligns with previous findings that Qwen-VL-Plus's safety mechanisms are relatively weaker [Ying; arXiv 2024].
>
> ### 7. Details of adaptive cropping (AC).
>
> **We have introduced adaptive cropping is in Appendix B.3 of the manuscript.** we use GPT-4o-Mini to locate text boxes and crop the remaining parts. It outperforms center/random cropping by more effectively eliminating textual content. We'll add this explanation to the Implementation Details and the Ablation Study section.
>
> ### 8. Why Warning Prompt work
>
> We hypothesize that the warning prompt enhances safety awareness, making the model more cautious with deceptive queries common in the dataset, improving overall performance. See our case study in Figure 2 (https://anonymous.4open.science/r/ICML2025-1-3F36/README.md).
> This observation aligns with prior defense works like [Zheng; arXiv 2025], which reports that explicit safety prompts can enhance baseline model accuracy.

---

### Decision · Program_Chairs · 2025-05-01

**Decision:**

Accept (poster)

**Comment:**

This work proposes DPS (Defense through Partial-Perception Supervision), a black-box, training-free defense method for Large Vision Language Models (LVLMs) that leverages responses from partial images to supervise the model's output. By prompting the model with partial-perception responses, DPS effectively reduces vulnerability to vision attacks while preserving accuracy on clean inputs. During the rebuttal phase, the authors have well addressed most reviewers' concerns. The Area Chair has discussed with reviewers and read through the submission and the author-reviewer response carefully and we decide together to accept this submission to ICML.